# Periodontitis, dental plaque, and atrial fibrillation in the Hamburg City Health Study

Julia Struppek[1]ᵒ, Renate B. Schnabel[2,3]ᵒ, Carolin Walther[1], Guido Heydecke[1], Udo Seedorf[1], Ragna Lamprecht[1], Ralf Smeets[4,5], Katrin Borof[1,6], Tanja Zeller[2,3], Thomas Beikler[7], Christin S. Börschel[2,3], Mahir Karakas[2,3], Martin Gosau[4], Ghazal Aarabi[1]*

1 Department of Prosthetic Dentistry, Center for Dental and Oral Medicine, University Medical Center Hamburg-Eppendorf, Hamburg, Germany, 2 Department of Cardiology, University Heart and Vascular Center, Hamburg, Germany, 3 German Center for Cardiovascular Research (DZHK e.V.) Partner Site Hamburg/Lübeck/Kiel, Munich, Germany, 4 Department of Oral and Maxillofacial Surgery, University Medical Center Hamburg-Eppendorf, Hamburg, Germany, 5 Division of "Regenerative Orofacial Medicine", Department of Oral and Maxillofacial Surgery, University Medical Center Hamburg-Eppendorf, Hamburg, Germany, 6 Epidemiological Study Center, University Medical Center Hamburg-Eppendorf, Hamburg, Germany, 7 Department of Periodontics, Preventive and Restorative Dentistry, University Medical Center Hamburg-Eppendorf, Hamburg, Germany

ᵒ These authors contributed equally to this work.
* g.aarabi@uke.de

**Data Availability Statement:** All relevant data are within the paper and its Supporting Information files.

## Abstract

### Background/Aim

Atrial fibrillation (AF) is a major health problem and causes heart failure and stroke. Pathophysiological mechanisms indicate a link with oral health including periodontitis (PD), but supporting data are scarce. The aim was to investigate the link between features of oral health and the prevalence of AF.

### Methods

This cross-sectional analysis of the Hamburg City Health Study included 5,634 participants with complete data on their PD and AF status. AF was assessed via self-reported questionnaire or medically diagnosed by standard 12-lead resting ECG. The oral health examination included full-mouth measurements of the dental plaque index (PI), the clinical attachment loss (CAL) at 6 sites per tooth, the bleeding on probing (BOP) and the decayed, missing and filled teeth (DMFT) index. Descriptive analyses for all variables stratified by the status of PD were performed. To test for an association between prevalent PD and prevalent AF, multivariable logistic regression models were used. Mediation analysis was used to test if interleukin-6 (IL-6) and/or C-reactive protein (CRP) mediated the association between PD and AF.

### Results

Atrial fibrillation (prevalence: 5.6%) and the severity of PD (prevalence: moderate: 57.7%, severe: 18.9%) increased with age in men and women. Prevalent severe PD, CAL ≥3 mm, PI, and BOP were all associated with prevalent AF in unadjusted regression analysis.

**Funding:** This study was funded in the form of grants by the Else Kröner-Fresenius-Foundation (DE) (Grant No 2017_A166) awarded to GA, and the European Research Council (Horizon 2020) (Grant No 648131 and 847770), the German Center for Cardiovascular Research (DE) (Grant No 81Z1710103), the German Ministry of Research and Education (DE) (Grant No 01ZX1408A), and the European Research Council (Horizon 2020, ERACoSysMed3) (Grant No 031L0239) awarded to RBS.

**Competing interests:** The authors have declared that no competing interests exist.

However, no association except for PI (odds ratio (OR): 1.22, 95% confidence interval (CI): 1.1–1.35, p<0.001) could be observed after adjusting for age, sex, high-sensitivity C-reactive protein (hs-CRP), interleukin-6 (IL-6), body mass index, diabetes, smoking, and educational level. Participants brushing their teeth at least twice daily had a lower AF prevalence compared with those brushing only once daily. Hs-CRP, IL-6, and the odds of AF increased as a function of PD severity grades in unadjusted analysis. However, neither the DMFT index nor IL-6 or CRP was associated with AF after adjusting for age and sex. Mediation analyses could not provide support for the hypothesis that IL-6 or CRP acted as mediator of the association between prevalent PD and prevalent AF.

## Conclusion

The study shows an association between prevalent AF and increased dental plaque levels indicated by a higher PI. In contrast, an association of prevalent PD with prevalent AF after adjustments for several confounders could not be demonstrated. Further studies are necessary to investigate the mechanisms underlying poor oral hygiene and AF as well as the influence of improved oral hygiene on AF onset.

## Introduction

Atrial fibrillation (AF) is the most frequent cardiac arrhythmia and one of the biggest health problems with a prevalence of 1–2.5% in Western countries [1, 2]. AF comes along with major complications like stroke and heart failure [3, 4]. Different risk factors such as age, sex, and cardiovascular disease have been established [5]. In recent years, the influence of inflammation on the occurrence of AF has become more evident and previous studies have shown a correlation between systemic inflammation and AF [6, 7]. According to Jalife et al. [8], the structural remodeling of the left atrium plays an important role in the pathophysiology of AF. However, not all signal cascades that trigger structural remodeling have been identified. Elevated levels of inflammatory biomarkers, like C-reactive protein (CRP), interleukin-6 (IL-6) and tumor necrosis factor α (TNF-α) [6, 9, 10] were observed in AF patients. Moreover, myocardial biopsies from AF patients showed inflammatory infiltrates in the atrial tissue [11].

A link between oral inflammation and several common diseases such as heart disease, stroke, and diabetes, has been observed [12–14]. Periodontitis (PD), which has a prevalence of 70% in German adults [15], is one of the most common oral disorders. The disease is primarily based on the dysbiosis in the accumulated biofilm [16]. Other co-factors are genetic predisposition (host response), smoking and various general diseases [17]. PD is characterized by a destruction of the supporting tissue of the teeth which causes gum bleeding recessions, and enlarged periodontal pockets resulting in tooth-loss if left untreated [18]. Several studies have confirmed increases of systemic inflammation markers in patients with PD [19, 20] and a reduction of serum inflammatory markers in PD patients after periodontal therapy was observed [21].

According to recently published results from the dental cohort of the Atherosclerosis Risk in Communities Study (ARIC), severe PD at baseline was associated with incident AF during follow-up and mediation analysis indicated that AF may mediate the association between PD and stroke [22]. Thus, the aim of the present study was to investigate whether an association between prevalent PD and prevalent AF could be demonstrated in the Hamburg City Health

Study (HCHS), which is a large-scale population-based cohort with comprehensive oral and cardiovascular health data, in order to provide further evidence for a relationship between both diseases.

## Materials and methods

### Subjects, study design, and setting

The Hamburg City Health Study (HCHS) is an ongoing regional prospective cohort study launched in 2016 and performed by the University Medical Center Hamburg-Eppendorf [23]. The aim of the HCHS is to investigate major chronic diseases like stroke, AF, dementia, and their risk factors with an extensive baseline assessment including clinical examinations and self-reported questionnaires.

This study represents a cross-sectional analysis of data from the first 10,000 participants at baseline recruited between February 2016 and November 2018. The general inclusion criteria were: residence in Greater Hamburg, German language knowledge, aged between 45–74 years, and no abnormal medical history (current alcohol abuse or alcohol dependence, use of drugs or oral steroids, and headache due to severe head injury). Exclusion criteria were participants requiring endocarditis prophylaxis, missing periodontal examination and/or missing data for AF leaving 5,634 individuals for the analysis (see S1 Fig for a detailed description of the sample).

This investigation was performed in accordance with the Declaration of Helsinki with accepted ethical standards for research practice [24]. The study protocol was approved by the data protection officer of the University Medical Centre of the University of Hamburg-Eppendorf and the local data protection commissioner (reg.-no.: PV5131). All participants signed a written declaration of consent before registering and participating in the study, which was voluntary. The article was written in accordance with recommendations issued by the "The Strengthening the Reporting of Observational Studies in Epidemiology (STROBE) Initiative" [25].

### Assessment of dental variables

All included participants had received a full dental and oral examination (except the third molars). The examination was carried out by trained and certified study nurses, consisting of advanced dentistry students performing their DDS/DMD doctoral thesis work, especially trained dentist's assistants, and other medical assistants with extensive experience in conducting the examination, which was performed according to a pre-specified SOP under the supervision of a certified dentist. The "study nurses" collected the raw data, such as number of teeth, pocket depths, number of bleeding points on probing etc., which were then used by two certified dentists to establish the diagnosis. In cases of disagreement, consensus was established by consulting a third dentist. Data accuracy was established by training and calibration of the staff, electronic data capture and transfer, longitudinal performance evaluation, and statistical monitoring. The Kappa statistics for the tooth count assessment ranged from 0.96 to 1.00, for untreated dental caries Kappa scores were 0.93 to 1.00. The overall Kappa statistics for identifying more complex traits, such as moderate and severe periodontitis according to the CDC/AAP case definition, ranged between 0.60 and 0.70, which was comparable with similar published results by Dye et al. for NHANES [26]. At the beginning, oral hygiene habits, antibiotic intake and a need for endocarditis prophylaxis were queried. Based on the measurements, a calculation of the decayed, missing and filled teeth (DMFT) index was performed and the plaque index (PI) of Silness-Löe (1964) [27] was assessed.

In order to assess the periodontal health, the probing depth of the gum pocket and the gingival (gum) recession were measured at 6 sites per tooth (mm) with a PCP-12 measuring probe. The clinical attachment loss (CAL) (mm) was calculated based on the measurement of probing depth and gingival recession. Additionally, the bleeding-on-probing (BOP) index (yes/no per tooth, expressed in % of bleeding sites) was measured. The diagnosis of PD and the classification of severity were performed based on the criteria of Eke and Page [28]:

- none/mild periodontitis: ≥2 interproximal sites with attachment loss (AL) ≥3 mm, and ≥2 interproximal sites with probing depth ≥4 mm (not on same tooth) or one site with probing depth ≥5 mm

- moderate periodontitis: ≥2 interproximal sites with AL ≥4 mm (not on same tooth), or ≥2 interproximal sites with probing depth ≥5 mm (not on same tooth)

- severe periodontitis: ≥2 interproximal sites with AL ≥6 mm (not on same tooth) and ≥1 interproximal site with probing depth ≥5 mm

### Assessment of cardiac variables

Demographic (age and sex) and anamnestic data were collected in advance by means of questionnaires. Based on the participant's medical history and a 12-lead resting ECG, AF was diagnosed (self-reported and medically diagnosed AF). Diagnosis and data were interpreted by two experienced cardiologists and in cases of disagreement reviewed by a third cardiologist.

### Assessment of biomarkers

For the biomarker analyses of high-sensitivity C-reactive protein (hs-CRP) and high-sensitivity interleukin-6 (hs-IL-6), venous blood samples were obtained from each participant. Hs-CRP was measured in the central lab of the UKE by routine measurements. The blood sample for IL-6 was frozen temporarily in tubes at—80˚C for storage. After thawing, 0.2 ml of plasma was analyzed using the Human IL-6 Quantikine HS ELISA Kit (High Sensitivity) (sensitivity: 0.11 pg/mL assay range: 0.2–10 pg/mL).

### Statistical analysis

Descriptive analyses were performed for all variables stratified by the status of PD. The frequency distributions of the categorical variables and medians including interquartile ranges (IQRs) for continuous variables were calculated. For bivariate analyses, p-values were calculated by the chi-square test for categorical variables and by a Kruskal-Wallis test for continuous variables. In order to test for associations between oral health measures and AF, multivariable logistic regression analyses were used. Three models were constructed: 1) bivariate unadjusted model, 2) adjusted for age and sex, 3) adjusted for several potentially relevant confounders (age, sex, body mass index, diabetes, smoking, educational level, and hs-CRP). Odds ratios (OR) with their 95% confidence intervals (CI) were reported. Values of $p < 0.05$ were considered significant. Statistical analyses were performed with the RStudio Version 1.1.453 for Windows.

### Statistical power considerations

Considering a sample size of n = 5,000, a prevalence of 25% for severe and of 30% for no or mild PD in older German people [15], a prevalence of AF of 6% [1], and assuming a true

unadjusted odds ratio for exposed subjects relative to unexposed subjects of 1.5, the sample size is sufficient to reject the null hypothesis that this odds ratio equals 1 with a power of 81% (type I error probability: 0.05; statistical model: Continuity-corrected $chi^2$ statistic).

## Results

### Baseline characteristics

The baseline characteristics of the cohort are summarized in Table 1 stratified by PD severity. Of participants with none/mild PD, 60.4% were women, 50.7% with moderate PD and 39.1% with severe PD. The severity of PD increased with age (median age in years [IQR]): none/mild PD 59 [52, 66]; moderate periodontitis 63 [55, 69]; severe periodontitis 66 [59, 71]). The prevalence of hypertension (72.5% in severe PD vs. 54.8% in none/mild PD) and of heart failure (2.7% in severe PD vs. 1.6% in none/mild PD) were higher in patients with severe PD compared with none/mild PD. AF was elevated in patients with severe PD (6.9%) in comparison to none/mild PD (4.3%). The proportion of patients affected by myocardial infarction or stroke was highest in the severe PD group. Median levels of inflammatory markers were higher in people with severe PD and the PI, number of teeth loss, and the BOP index increased according to PD severity (Table 1).

### Subgroup results of periodontitis

Table 2 presents baseline characteristics stratified by age and sex. The prevalence of severe PD increased with age in both sexes (8.4% of the 45–54 years and 19.7% of the 65+ aged female participants suffered from severe PD). Male study participants were more often affected by PD than women: the severe PD group included 22.2% of the 55-64-year-old and 29.1% of the 65 + year-old male participants.

### Subgroup results of atrial fibrillation

Fig 1 shows an increased prevalence of AF with increasing age in both sexes. The prevalence of AF in the group from 45–54 years was low. In men, 13.2% of the age group 65+ had AF, whereas AF was less prevalent in women of the same age group (8.5%). AF was slightly more prevalent in men aged 55–64 years who suffered from moderate/severe PD.

### Interaction between AF and oral hygiene habits, plaque index and periodontitis

Participants aged ≥55 years who cleaned their teeth only once a week were more frequently affected by AF in comparison to patients with daily/twice a day cleaning frequencies (Fig 2).

### Multiple logistic regression models

Significant associations of prevalent severe PD, PI, and BOP with prevalent AF were observed in the bivariate unadjusted models. After adjusting for age, sex, hs-CRP, BMI, diabetes, smoking, and educational level, the relationships of AF with severe PD and BOP were no longer significant. In contrast, the PI was significantly associated with AF after these adjustments (p <0.001) (Table 3). As shown in Table 4, the DMFT index was not associated with AF after adjustment for age and sex.

**Table 1. Characteristics of the study population stratified by grades of periodontitis.**

| | Degree of Periodontitis | | |
|---|---|---|---|
| | **None/mild** | **Moderate** | **Severe** |
| | **N = 1453** | **N = 3580** | **N = 1176** |
| | Median [IQR] or N (%) | | |
| Socio-demographic characteristics | | | |
| Sex = Female | 878 (60.4) | 1814 (50.7) | 460 (39.1) |
| Age | 59 [52, 66] | 63 [55, 69] | 66 [59, 71] |
| Cardiovascular risk factors | | | |
| BMI | 25.56 [23.01, 28.67] | 26.02 [23.55, 29.01] | 26.45 [24.11, 29.65] |
| Smoking | | | |
| Current | 235 (16.2) | 608 (17.1) | 293 (25.1) |
| Former | 625 (43.2) | 1581 (44.4) | 550 (47) |
| Never | 588 (40.6) | 1369 (38.5) | 326 (27.9) |
| Diabetes = Yes | 85 (6.2) | 242 (7.4) | 122 (11.3) |
| Hypertension = Yes | 768 (54.8) | 2266 (66.3) | 810 (72.5) |
| Systolic blood pressure | 134 [123.5, 146.5] | 137.5 [125.5, 150.5] | 139.5 [127.5, 153] |
| Heart parameter | | | |
| Heart rate | 67.5 [61, 75] | 68.5 [62, 76] | 68.5 [61.5, 76.5] |
| LVEF | 58.49 [54.58, 62.98] | 58.57 [54.56, 62.67] | 57.99 [53.45, 62.15] |
| Heart failure = Yes | 23 (1.6) | 76 (2.1) | 31 (2.7) |
| Atrial fibrillation = Yes | 57 (4.3) | 181 (5.5) | 75 (6.9) |
| Myocardial infarction = Yes | 31 (2.1) | 88 (2.5) | 38 (3.3) |
| Stroke = Yes | 27 (1.9) | 98 (2.8) | 45 (3.9) |
| hs-CRP | 0.1 [0.06, 0.23] | 0.11 [0.06, 0.25] | 0.13 [0.07, 0.3] |
| hs-IL-6 | 1.47 [1.03, 2.08] | 1.57 [1.16, 2.23] | 1.8 [1.34, 2.69] |
| Dental parameter | | | |
| Cleaning teeth | | | |
| Never | 3 (0.2) | 1 (0) | 1 (0.1) |
| Once per week | 1 (0.1) | 6 (0.2) | 1 (0.1) |
| Once per day | 172 (12.4) | 425 (12.5) | 160 (14.5) |
| 2x per day | 1214 (87.3) | 2969 (87.3) | 941 (85.3) |
| DMFT-Index | 17 [14, 21] | 19 [16, 23] | 21 [17, 24.25] |
| Number of missing teeth | 2 [0, 4] | 2 [1, 5] | 4 [1, 9] |
| BOP | 2.08 [0, 7.14] | 8.33 [2.17, 19.23] | 21.05 [9.26, 41.67] |
| Plaque index | 0 [0, 10.71] | 8.93 [0, 27.78] | 22 [5.77, 54.76] |

Abbreviations: BMI, body mass index; LVEF, left ventricular ejection fraction; hs-CRP, high-sensitivity C-reactive protein; hs-IL-6, high-sensitivity interleukin-6; DMFT-Index, decayed, missing and filled teeth index; BOP, bleeding on probing

## Subgroup analyses in participants with hypertension and participants lacking risk factors

To study the potential association between prevalent PD and AF in low and high risk groups, we performed multivariable regression analyses in participants with a history of hypertension (high risk group, n = 3,844) and a second group without a history of hypertension, stroke, diabetes mellitus, heart failure, myocardial infarction, current smoking, and/or BMI above 35 (low risk group, n = 1,319, 18 cases of AF, prevalence: 1.36%). As shown in S1 Table no association was observed based on the crude logistic regression model without any adjustments in

**Table 2. Periodontal status, plaque index, BOP and AF related to age decades.**

| | Population | | | | | | |
|---|---|---|---|---|---|---|---|
| | Male | | | Female | | | |
| | **45–54** | **55–64** | **65+** | **45–54** | **55–64** | **65+** | |
| | N = 1046 | N = 1577 | N = 2269 | N = 1227 | N = 1745 | N = 2136 | |
| | Median [IQR] or N (%) | | | | | | |
| Periodontitis | | | | | | | < 0.001 |
| None/mild | 199 (29.7) | 183 (18) | 193 (14.1) | 306 (38.9) | 311 (28.1) | 261 (20.7) | |
| Moderate | 380 (56.6) | 610 (59.9) | 776 (56.8) | 415 (52.7) | 650 (58.7) | 749 (59.5) | |
| Severe | 92 (13.7) | 226 (22.2) | 398 (29.1) | 66 (8.4) | 146 (13.2) | 248 (19.7) | |
| Plaque Index | 8.7 [0, 26.92] | 10.87 [0, 34] | 17.39 [1.85, 46.86] | 3.7 [0, 17.86] | 4.35 [0, 1.43] | 8.33 [0, 6.96] | < 0.001 |
| BOP | 7.14 [1.92, 17.35] | 8.93 [2, 23.04] | 9.26 [2.38, 22] | 5.77 [1.79, 17.31] | 7.41 [1.85, 9.64] | 8.33 [1.98, 20.09] | < 0.001 |
| AF = Yes | 10 (1) | 62 (4.3) | 276 (13.2) | 9 (0.8) | 42 (2.7) | 162 (8.5) | < 0.001 |

Abbreviations: BOP, bleeding on probing; AF, atrial fibrillation; IQR, interquartile range.

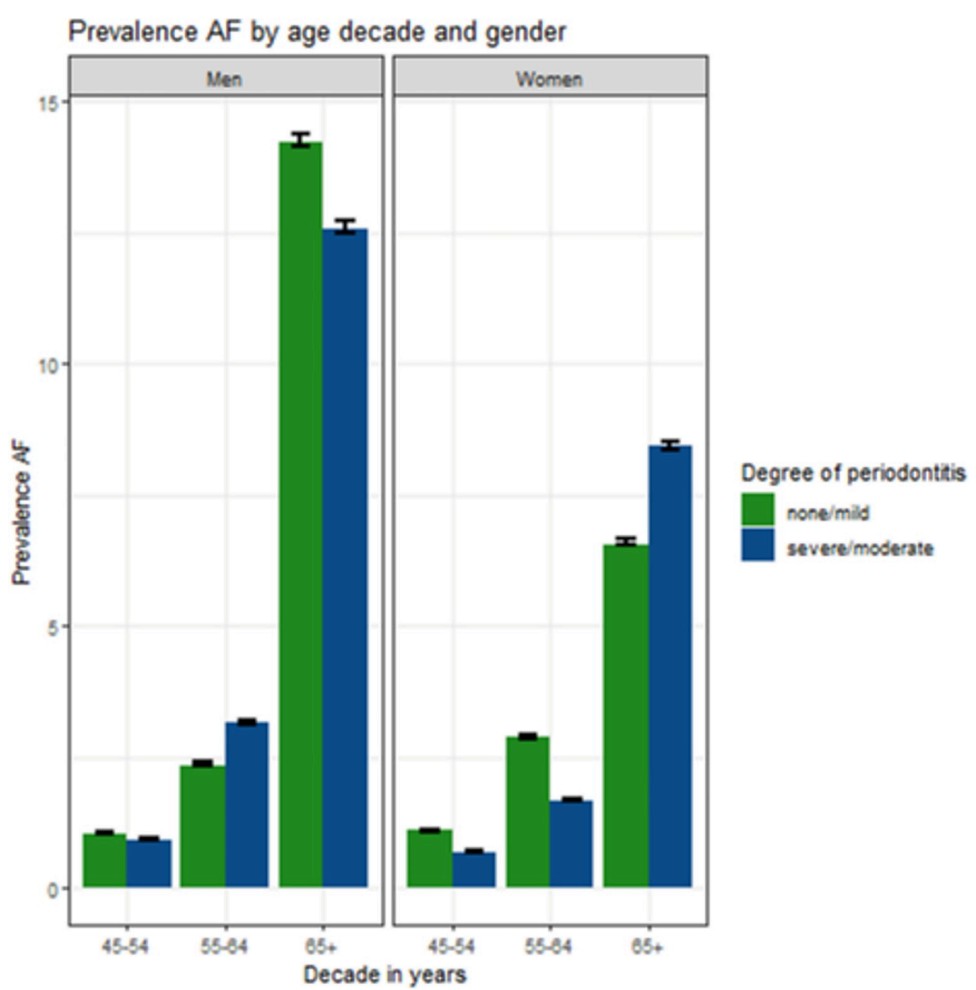

**Fig 1. Prevalence of AF (atrial fibrillation) for females and males in different age decades (45–54, 55–64, 65+) classified according to the degrees of periodontitis none/mild and severe/moderate.**

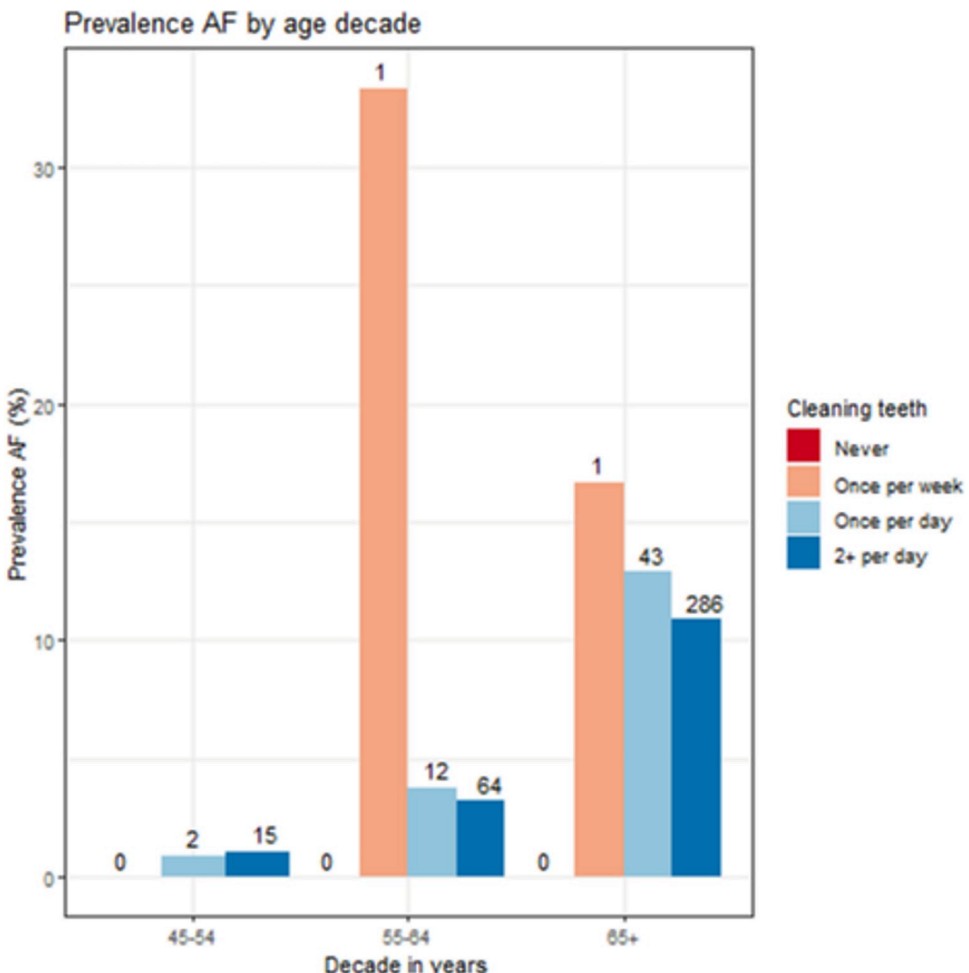

**Fig 2. Prevalence of AF (atrial fibrillation) (N, %) for ascending age decades (45–54, 55–64, 65+ years) and different oral hygiene habits.** Cleaning frequencies: Never, once per week, once per day, twice per day).

the low risk group. The age- and sex adjusted multivariable logistic regression model for the low risk group, which is shown in S2 Table, shows that age was strongly associated with AF, whereas sex was not associated with AF. Similar results were observed in the high risk group except that both, age and sex, were associated with AF (S3 and S4 Tables).

**Table 3. Logistic regression models (outcome: AF).**

| | Periodontal parameters | | | | | | Periodontitis | | | |
|---|---|---|---|---|---|---|---|---|---|---|
| | CAL ≥ 3 mm | | Plaque index | | BOP | | Moderate | | Severe | |
| | OR (CI) | p-value | OR (CI) | p-value | OR (CI) | p-value | OR (CI) | p-value | OR (CI) | p-value |
| Model 1 | **1.19 (1.07–1.33)** | **0.002** | **1.36 (1.25–1.48)** | **<0.001** | **1.11 (1.00–1.23)** | **0.046** | 1.31 (0.96–1.77) | 0.085 | **1.66 (1.16–2.36)** | **0.005** |
| Model 2 | 0.99 (0.88–1.12) | 0.91 | **1.19 (1.09–1.30)** | **<0.001** | 1.07 (0.96–1.19) | 0.24 | 0.93 (0.68–1.27) | 0.64 | 0.92 (0.64–1.33) | 0.67 |
| Model 3 | 1.04 (0.91–1.18) | 0.58 | **1.22 (1.10–1.35)** | **<0.001** | 1.07 (0.95–1.20) | 0.28 | 0.93 (0.66–1.42) | 0.70 | 0,94 (0.63–1.42) | 0.78 |

Bold: p<0.05. Abbreviations: hs-CRP, high-sensitivity C-reactive protein; BMI, body mass index; CAL, clinical attachment loss; BOP, bleeding on probing; OR, odds ratio; CI, confidence interval.

Adjustment: Unadjusted (Model 1), adjusted for age, sex (Model 2) and for age, sex, log (hs-CRP), BMI, diabetes, smoking, and education (Model 3).

Table 4. Association between the DMFT index (exposure) and AF (outcome).

| Predictors | OR (CI) | p-value |
|---|---|---|
| DMFT-index | 1.00 (0.98–1.02) | 0.854 |
| **Age** | **2.54 (2.25–2.88)** | **<0.001** |
| **Sex (women)** | **0.63 (0.52–0.76)** | **<0.001** |

Bold: p<0.05. Abbreviations: DMFT, decayed-missing-filled-teeth; OR, odds ratio; CI, confidence interval.

## Mediation analyses for CRP and IL-6

CRP, IL-6, and the odds of AF increased as a function of PD severity grades in unadjusted analysis (S5 Table). However, the estimates for the average causal mediation effects (ACME) observed for IL-6 and hs-CRP were not significant (S6 Table). The average direct effect (ADE) reached the level of significance for hs-CRP (p = 0.046) but was very small and negative.

## Discussion

The cross-sectional data of the population-based cohort study shows associations between several measures of oral health, such as CAL, PI, severe PD, and BOP, and prevalent AF. All but PI lost statistical significance after adjustment for common confounders. The findings thus indicate that the observed associations of oral health parameters with prevalent AF are largely explained by sociodemographic, particularly age and sex, in this sample. This is also supported by the subgroup analyses on hypertensive participants and on participants lacking risk factors. The crude logistic regression model without any adjustments resulted in p-values clearly above 0.05, suggesting that an association between PD and AF was unlikely. However, it should be noted that due to the low prevalence of AF, the power to detect an association in the subgroup analyses may have been insufficient.

The findings are in agreement with results from a cross-sectional study on Danish octogenarians published by Holm-Pedersen et al. who observed an association between root caries and cardiac arrhythmias, but found no association between PD and AF [29]. Conversely, Im et al. observed a higher incidence of arrhythmias including AF in the PD group of a much smaller prospective study [30]. The median age of the non-PD group was below our median age and above in the PD group. Recently published results from the dental cohort of the ARIC study demonstrated that severe PD at baseline was associated with incident AF during follow-up in subjects without AF at baseline. In addition, mediation analysis indicated that AF may mediate the association between PD and stroke [22]. Mean follow-up for AF was over 17 years and Cox proportional hazards models adjusted for AF risk factors were used. The ARIC observations relate to a prospective analysis of AF incidences occurring during a long follow-up period in subjects without AF at baseline, whereas our observations relate to a cross-sectional analysis of the prevalences of PD and AF at baseline with no follow-up data yet available. Since PD and AF strongly correlate with age [15, 31], these divergent results may likely be explained by the differences of the study designs, differing impact of confounders, and/or different age ranges of the study samples.

We relied on a combination of self-reporting and ECG recordings to diagnose AF because paroxysmal and some persistent forms of AF are difficult to detect by a single ECG examination in an epidemiological study. Therefore, we included also the medical history to identify such cases. People are usually aware of their diagnosis and about ¼ of participants who reported a history of AF could actually be verified by ECG. In contrast, only 0.4% of those who reported no history of AF presented with AF according to the ECG. This shows that verifiable

cases were highly enriched in the self-reporting group. Nevertheless, it cannot be excluded that including the self-reported cases may have led to some miss-classification. On the other hand, some cases of paroxysmal or persistent AF would have been missed, if we had decided to not consider the reports. Another limitation relates to the relatively low power of the subgroup analyses, which have therefore to be interpreted prudently.

Our data show a significant association between plaque accumulation and prevalent AF that is independent of age, sex, hs-CRP, BMI, smoking, diabetes, and educational status. This supports the hypothesis that dental plaque and the associated acute inflammatory response may have an impact on the development of AF, stronger than the BOP or PD as defined by clinical attachment loss. Various studies have already shown possible links between acute and/or chronic inflammation and AF (reviewed in ref. [32]). Higher systemic pro-inflammatory activity due to oral inflammation reflected by plaque burden may promote atrial remodeling and trigger AF. A potential contribution of *Porphyronomas gingivalis*, which is one of the key periodontitis-associated bacteria, to the risk of AF has been proposed [33]. *P. gingivalis* and its virulence factors as well as inflammatory mediators could be involved in cardiac remodeling. In our study, CRP, IL-6, and the odds of AF increased as a function of PD severity grades in non-adjusted analysis. Thus, our data do not exclude that the inflammatory markers play a role in the mechanism of AF. The statistical models show that neither the DMFT nor IL-6 or CRP is associated with AF after adjusting for age and sex, whereas aging and male sex are strongly associated with prevalent AF. It is known that the plasma concentrations of CRP and IL-6 are higher in healthy individuals aged over 65 years compared to younger people [34] and that healthy women have higher CRP levels than healthy men [35]. On the other hand, aging and sex are linked to a vast number of additional effects. Therefore, an investigation of the causal relationship between dental plaque accumulation and AF is beyond the scope of the study.

That oral hygiene may play a role in the prevention of AF is supported by the recent ARIC data [22] and by studies published by Chen et al. and Chang et al. who showed that the removal of biofilms by dental scaling or improving oral hygiene habits could reduce the risk of AF [36, 37].

In conclusion, this study shows an association between dental plaque accumulation and AF which was independent of the tested confounders. Further studies are necessary to investigate the relationship between oral inflammation and AF as well as the impact of improved oral hygiene on the prevention of AF.

## Supporting information

**S1 Fig. Sampling chart.** 9,087 ECGs, 9,895 medical histories, and 6,209 full-mouth periodontal examinations were evaluated for the study. Included were 5,634 participants with complete PD and AF classification data. The AF prevalence was 5.6% in the subgroup with complete data compared to 5.7% in the subgroup with AF data. Of 169 AF cases, which were identified by ECG, 129 cases (75%) had a positive medical history of AF.
(PDF)

**S1 Table. Association between periodontitis and atrial fibrillation in low risk subjects: Crude logistic regression model.** The potential association between PD and AF was studied in a low risk group. Participants with a history of hypertension, stroke, diabetes mellitus, heart failure, myocardial infarction, current smoking, and/or BMI above 35 (N = 4,890) were excluded. This resulted in 1,319 subjects, which included 18 cases of AF (AF prevalence: 1.36%). Shown are the results of the crude logistic regression model without any adjustments.
(DOCX)

**S2 Table. Association between periodontitis and atrial fibrillation in low risk subjects: Multivariable logistic regression model.** Age was strongly associated with AF, which corresponded with the results shown in the main paper for the whole sample of 5,634 participants. In contrast, sex was not associated with AF, which differed from the results obtained for the whole sample, in which women had lower odds of AF than men.
(DOCX)

**S3 Table. Arterial hypertension and periodontitis severity grades.** In order to examine the association between PD and AF in a high risk group, subgroup analysis in participants with hypertension was performed. As shown in the Table, 3,844 participants had hypertension and the fraction of participants with severe PD was higher in hypertensive compared with normotensive participants.
(DOCX)

**S4 Table. Association between periodontitis and atrial fibrillation in hypertensive subjects: Multivariable logistic regression model.** The Table shows the age- and sex adjusted multivariable logistic regression model for the high risk group. Age and sex were strongly associated with AF, which corresponded with the results shown in the main paper for the whole sample of 5,634 participants. However, no association between severe or moderate PD and AF could be observed.
(DOCX)

**S5 Table. CRP and IL-6 plasma concentrations according to periodontitis grades.** CRP, IL-6, and the odds of AF increased as a function of PD severity grades in non-adjusted analysis.
(DOCX)

**S6 Table. Mediation analysis for IL-6 and CRP.** Exposure: Periodontitis, outcome: Atrial fibrillation.
(DOCX)

## Author Contributions

**Conceptualization:** Udo Seedorf, Ralf Smeets, Ghazal Aarabi.

**Data curation:** Julia Struppek, Renate B. Schnabel, Ragna Lamprecht, Christin S. Börschel.

**Formal analysis:** Katrin Borof.

**Funding acquisition:** Renate B. Schnabel, Guido Heydecke, Udo Seedorf, Ghazal Aarabi.

**Methodology:** Renate B. Schnabel, Carolin Walther, Udo Seedorf, Katrin Borof, Tanja Zeller, Thomas Beikler, Mahir Karakas, Martin Gosau, Ghazal Aarabi.

**Supervision:** Guido Heydecke, Udo Seedorf, Tanja Zeller, Ghazal Aarabi.

**Validation:** Christin S. Börschel, Mahir Karakas, Ghazal Aarabi.

**Writing – original draft:** Julia Struppek.

**Writing – review & editing:** Renate B. Schnabel, Carolin Walther, Guido Heydecke, Udo Seedorf, Ragna Lamprecht, Ralf Smeets, Katrin Borof, Tanja Zeller, Thomas Beikler, Mahir Karakas, Martin Gosau, Ghazal Aarabi.

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
