## [Editor Report · Decision Letter 0]

26 Mar 2021

PONE-D-21-08970

Oral health and atrial fibrillation in the Hamburg City Health Study  

PLOS ONE

Dear Dr. Seedorf,

Thank you for submitting your manuscript to PLOS ONE. This is an interesting topics. Before sending out for external review, I would like to clarify some points. You may remove the patients with AF risk (i.e., HTN, DM, Stroke events, etc.) from the analyses. Then, you may see clear results without the statistical models.  Please analyze h-CRP and IL-6 as well. If the inflammatory markers do not play a role in AF, what would be potential mechanism? In addition, I do not think reference 25 would be a suitable to cite since these studies are  merely based upon insurance claims, not clinical data.

We look forward to receiving your revised manuscript.

Kind regards,

Tomohiko Ai, M.D., Ph.D.

Academic Editor

PLOS ONE

Additional Editor Comments:

This is an interesting topics. Before sending out for external review, I would like to clarify some points. You may remove the patients with AF risk (i.e., HTN, DM, Stroke events, etc.) from the analyses. Then, you may see clear results without the statistical models. Please analyze h-CRP and IL-6 as well. If the inflammatory markers do not play a role in AF, what would be potential mechanism?

Journal Requirements:

Please include your tables as part of your main manuscript and remove the individual files. Please note that supplementary tables should be uploaded as separate "supporting information" files.

---

## [Author Response · Author response to Decision Letter 0]

18 May 2021

Please, see the attached "Response to Reviewers" file

---

## [Decision Letter · Decision Letter 1]

7 Jul 2021

PONE-D-21-08970R1

Oral health and atrial fibrillation in the Hamburg City Health Study  

PLOS ONE

Dear Dr. Seedorf,

Thank you for submitting your manuscript to PLOS ONE. After careful consideration, we feel that it has merit but does not fully meet PLOS ONE’s publication criteria as it currently stands. Therefore, we invite you to submit a revised version of the manuscript that addresses the points raised during the review process

Your paper was reviewed by three experts in the field. Although the topic seems to be interesting, several issues need to be clarified. Specifically, there are several confound factors need to be included to asses the PD status, and occurrence of AF needs to be validated. Please read the comments carefully, and address the issues accordingly. I would recommend the authors to consult statistical expert(s).

We look forward to receiving your revised manuscript.

Kind regards,

Tomohiko Ai, M.D., Ph.D.

Academic Editor

PLOS ONE

Reviewers' comments:

Reviewer's Responses to Questions

**Comments to the Author**

1. If the authors have adequately addressed your comments raised in a previous round of review and you feel that this manuscript is now acceptable for publication, you may indicate that here to bypass the “Comments to the Author” section, enter your conflict of interest statement in the “Confidential to Editor” section, and submit your "Accept" recommendation.

Reviewer #1: (No Response)

Reviewer #2: (No Response)

Reviewer #3: (No Response)

2. Is the manuscript technically sound, and do the data support the conclusions?

Reviewer #1: Partly

Reviewer #2: Yes

Reviewer #3: No

3. Has the statistical analysis been performed appropriately and rigorously? 

Reviewer #1: Yes

Reviewer #2: Yes

Reviewer #3: No

4. Have the authors made all data underlying the findings in their manuscript fully available?

Reviewer #1: Yes

Reviewer #2: Yes

Reviewer #3: No

5. Is the manuscript presented in an intelligible fashion and written in standard English?

Reviewer #1: Yes

Reviewer #2: Yes

Reviewer #3: Yes

6. Review Comments to the Author

Reviewer #1: I find it hard to justify the accuracy of the data collected without more information- Therefore I feel the article can be accepted if they provide more information about the data collected.

It would be useful if the authors shared data on how many patients were already diagnosed with AF, and how many they tested for AF who were diagnosed with unknown AF. Without this data it is hard to see a true representative of data- e.g the patient may have had good plaque control, was diagnosed with AF but since the diagnosis had then had not brushed teeth as much and resulted in poor plaque control. In a cross sectional study ideally the patient would not know they have AF, have poor plaque control and after investigation with ECG, has been diagnosed with unknown AF. If a patient was already diagnosed with AF in the medical was this information known to the researcher prior to the plaque index being carried out? This can introduce researcher bias. Also there is no cause or effect with this type of study as it is cross-sectional they cannot state oral health preventing AF based on their study. If there was an exclusion criteria for those who already had AF then can this be stated clearly in the methods

I have some other comments:

According to a recently published systematic review, only two small clinical studies had been looked at a potential association between PD and AF over the last fifty years [22].

Incorrect, this SR states two small clinical studies looking at acute infections, not PD, also here is a useful and recent citation

Sen S, Redd K, Trivedi T, Moss K, Alonso A, Soliman EZ, Magnani JW, Chen LY, Gottesman RF, Rosamond W, Beck J, Offenbacher S. Periodontal Disease, Atrial Fibrillation and Stroke. Am Heart J. 2021 May;235:36-43. doi: 10.1016/j.ahj.2021.01.009. Epub 2021 Jan 24. PMID: 33503409; PMCID: PMC8084947.

The examination was carried out by trained and certified study nurses.

Normally dentists are required to examine patients and diagnose PD, not nurses- if nurses then is diagnosis inaccurate? If nurses are trained then how are they trained to ensure accurate diagnosis are they performing the periodontal charts? more detail needed here.

Based on the measurements, a calculation of the decayed, missing and filled teeth (DMFT) index was performed and the plaque index (PI) of Silness-Löe (1964) was assessed

Would have liked to see more information in the methods about systematic and rigourous testing of scores- for example, were the plaque scores assessed by 2 different clinicians, if there was disagreement then would a third clinician be involved

Although (DMFT) index is a valid measurement tool it has disadvantages as people can lose teeth from trauma or for orthodontic treatment or tooth decay (eg not just from periodontitis).

The diagnosis of PD and the classification of severity were performed based on the criteria of Eke and Page [26]:

26. Chen SJ, Liu CJ, Chao TF, Wang KL, Chen TJ, Chou P, et al. Dental scaling and atrial fibrillation: a 241 nationwide cohort study. International journal of cardiology. 2013;168(3):2300-3. doi: 242 10.1016/j.ijcard.2013.01.192. PubMed PMID: 23453452.

Incorrect citation placed.

Assessment of cardiac variables

Authors have used a combined method of diagnosing for AF including self reporting and medical investigations with 24 hour ECG- 1) self reporting is considered inaccurate for diagnosing AF, diagnosing all patients medically would have been more accurate

Methods

would like to see a flowchart of patient journey and data which would make the results of the research more easily accessible and clear

Statistical analysis

adjusted for all confounders considered relevant (age, sex, body mass index, diabetes, smoking, and hs-CRP)

There are many more confounders considered relevant- please amend statement to adjusted for some confounders considered relevant

Figure 1 and figure 2 are not clearly labelled, figure 1 no label of unit measurement for y axis is it a percentage, figure 2 for me was not absolutely clear what the units of measurements were as the bar charts had numbers on the top but I do not know what they represented.

Reviewer #2: The authors present a cross-sectional study evaluating the relation between PD and AF. Patients were evaluated for PD using CAL, BOP, DMFTi and PI. AF was assessed using a 24-hour ECG monitoring or a single 12 lead ECG, according to risk, using the CHARGE-AF score. In addition, diagnosis was made using self-reported AF or medically diagnosed AF. Their results reflect what we already know about the epidemiologic patterns of AF, it increases with age and other risk factors such male gender and hypertension.

Overall, after adjustment for multiple confounders, there was no association between PD and AF; however, the authors found an association between markers of oral hygiene like plaque index and AF. They conclude that there might be an association between dental plaque accumulation and AF.

The article is interesting as it adds to the evidence of the link between PD and CVD; particularly the relationship between PD and AF still needs to be confirmed. The text is written in good English, methods and statistical analysis are clear, and results are presented appropriately.

Revisions

1. Could the authors explain the rational for using the CHARGE-AF score to decide if patients where having a 24-hrs ECG monitoring or not in the methods section.

2. Add the CHARGE-AF score for each group of PD as a part of the results.

3. The authors include the LVEF in different groups of PD as a part of their results. Do they have information of other echocardiographic parameters? It would be particularly interesting if they provide data on left atrium (volume, diameter), since it is a marker of FA risk.

4. This is a suggestion: Most of the data that shows an association between PD and cardiac structural and functional abnormalities detected with echocardiogram (mass, myocardial deformation), comes from studies in patients with diabetes mellitus, hypertension and chronic kidney disease. In all of these studies, there was a clear association between PD severity and left ventricular hypertrophy and abnormal ventricular function. (You can see: Periodontal Disease, Systemic Inflammation and the Risk of Cardiovascular Disease. Heart Lung Circ. 2018 Nov;27(11):1327-1334, for a description of some of these studies). In our laboratory, we have done some work in otherwise healthy subjects with different grades of PD using echocardiography, and we could not find an association between PD and LVH or abnormalities in LV function. Although the reasons can be diverse, there is a possibility that PD and the systemic inflammatory process that comes with it acts as a boost for CVD; perhaps, PD might not be enough to cause CV abnormalities, but when added to other types of stress to the CV system (Hypertension, DM) it can cause CVD.

Since AF is multifactorial, and abnormalities in cardiac structure and function can play a role, this can be important for your study. More than 50 of your study population have hypertension; perhaps, you can consider a last analysis including only those with HTN and divide them in those with none/mild PD and severe PD to look for an association between AF and PD, in unadjusted and adjusted analysis.

Reviewer #3: 1) Recent results from ARIC cohort study on PD-AF association not discussed

2) DMFT assessed, a measure primarily for Caries. Data not used for analysis

3) CRP and IL-6 are potential mediators, hence should not be adjusted for

4) Socioeconomic status is a potential confounder that is not assessed

5) Based on hypothesis oral care should be associated with less AF

6) Assessment of AF using questionnaire not a validated method

7. PLOS authors have the option to publish the peer review history of their article (what does this mean?). If published, this will include your full peer review and any attached files.

Reviewer #1: **Yes: **Amaar Hassan

Reviewer #2: **Yes: **Edgar Francisco Carrizales-Sepúlveda

Reviewer #3: No

---

## [Author Response · Author response to Decision Letter 1]

2 Sep 2021

6. Review Comments to the Author

Reviewer #1: 

It would be useful if the authors shared data on how many patients were already diagnosed with AF, and how many they tested for AF who were diagnosed with unknown AF.

Rely: Of the 561 AF cases, 521 were already diagnosed with AF and 40 additional cases with previously unknown AF were newly identified in the study by the ECGs. The total number of ECGs was 9,087. We added a sampling flow chart to the supplementary material to better characterize the sample.

If a patient was already diagnosed with AF in the medical was this information known to the researcher prior to the plaque index being carried out?

Reply: No, this was not known.

There is no cause or effect with this type of study as it is cross-sectional they cannot state oral health preventing AF based on their study.

Reply: That is true. We removed the statement from the conclusions.

If there was an exclusion criteria for those who already had AF then can this be stated clearly in the methods

Reply: No, this was not an exclusion criterion. This is a cross-sectional study looking at prevalent AF and prevalent PD at baseline with PD as independent variable and AF as dependent variable.

According to a recently published systematic review, only two small clinical studies had been looked at a potential association between PD and AF over the last fifty years [22].

Incorrect, this SR states two small clinical studies looking at acute infections, not PD, also here is a useful and recent citation

Sen S, Redd K, Trivedi T, Moss K, Alonso A, Soliman EZ, Magnani JW, Chen LY, Gottesman RF, Rosamond W, Beck J, Offenbacher S. Periodontal Disease, Atrial Fibrillation and Stroke. Am Heart J. 2021 May;235:36-43. doi: 10.1016/j.ahj.2021.01.009. Epub 2021 Jan 24. PMID: 33503409; PMCID: PMC8084947.

Reply: Thank you. We discuss this excellent recently published work in the discussion of the revised manuscript. 

The examination was carried out by trained and certified study nurses.

Normally dentists are required to examine patients and diagnose PD, not nurses- if nurses then is diagnosis inaccurate? If nurses are trained then how are they trained to ensure accurate diagnosis are they performing the periodontal charts? more detail needed here.

Reply: The term “study nurse”, which is used here, refers to extensively trained and calibrated staff, including advanced dentistry students performing their DDS/DMD doctoral thesis work, especially trained dentist’s assistants, and other medical assistants with extensive experience in conducting the examination, which was performed according to a pre-specified SOP under the supervision of a certified dentist. The “study nurses” did not “diagnose periodontitis”, as you say. They collected the raw data, such as number of teeth, pocket depths, number of bleeding points on probing etc., which were then used by two certified dentists to establish the diagnosis. Data accuracy was established by training and calibration of the staff, electronic data capture and transfer, longitudinal performance evaluation, and statistical monitoring. The Kappa statistics for the tooth count assessment ranged from 0.96 to 1.00, for untreated dental caries Kappa scores were 0.93 to 1.00. The overall Kappa statistics for identifying more complex traits, such as moderate and severe periodontitis according to the CDC/AAP case definition, ranged between 0.60 and 0.70, which was comparable with published results by Dye et al. BMC Oral Health (2019) 19:95 for NHANES. Thus, we are confident that the data are accurate. On the other hand, the high fraction of missing data, which was due to a shortage of sufficiently well trained staff, should be noted as a disadvantage. We provided more details in the methods section and added a sampling chart to the supplementary material (Fig. S1). 

Based on the measurements, a calculation of the decayed, missing and filled teeth (DMFT) index was performed and the plaque index (PI) of Silness-Löe (1964) was assessed

Would have liked to see more information in the methods about systematic and rigourous testing of scores- for example, were the plaque scores assessed by 2 different clinicians, if there was disagreement then would a third clinician be involved

Reply: All scores that we used are in essence very simple and have been in use in dentistry already for a long time. We provided more details in the methods section.

 Although (DMFT) index is a valid measurement tool it has disadvantages as people can lose teeth from trauma or for orthodontic treatment or tooth decay (eg not just from periodontitis).

Reply: We agree with you. However, we have not used the DMFT for any evaluations. Nevertheless, the DMFT may be interesting for some readers because it is a widely used indicator of overall dental health. The actual values show that our cohort in fact has a very good oral health. 

The diagnosis of PD and the classification of severity were performed based on the criteria of Eke and Page [26]:

26. Chen SJ, Liu CJ, Chao TF, Wang KL, Chen TJ, Chou P, et al. Dental scaling and atrial fibrillation: a 241 nationwide cohort study. International journal of cardiology. 2013;168(3):2300-3. doi: 242 10.1016/j.ijcard.2013.01.192. PubMed PMID: 23453452.

Incorrect citation placed.

Reply: Thank you. We corrected this error. 

Assessment of cardiac variables

Authors have used a combined method of diagnosing for AF including self reporting and medical investigations with 24 hour ECG- 1) self reporting is considered inaccurate for diagnosing AF, diagnosing all patients medically would have been more accurate

Reply: Paroxysmal and some persistent forms of AF are difficult to detect by a single ECG examination in an epidemiological study. Therefore, we relied also on the medical history to identify such cases. People are usually aware of their diagnosis and about ¼ of participants who reported a history of AF could actually be verified by ECG. In contrast, only 0.4% of those who reported no history of AF presented with AF according to the ECG. This shows that verifiable cases were highly enriched in the self-reporting group. Nevertheless you are right that self-reporting may lead to some miss-classification. On the other hand, we would have missed some cases of paroxysmal or persistent AF, if we had decided to not consider the reports. We added this consideration to the discussion section.

 would like to see a flowchart of patient journey and data which would make the results of the research more easily accessible and clear

Reply: Very good suggestion. We added a flowchart to the supplementary material.

adjusted for all confounders considered relevant (age, sex, body mass index, diabetes, smoking, and hs-CRP)

There are many more confounders considered relevant- please amend statement to adjusted for some confounders considered relevant

Reply: Okay, done.

Figure 1 and figure 2 are not clearly labelled, figure 1 no label of unit measurement for y axis is it a percentage, figure 2 for me was not absolutely clear what the units of measurements were as the bar charts had numbers on the top but I do not know what they represented.

Reply: Please, see improved figures in the revised manuscript. 

Reviewer #2:

Could the authors explain the rational for using the CHARGE-AF score to decide if patients where having a 24-hrs ECG monitoring or not in the methods section.

Reply: This was originally planned according to the study protocol in order to identify cases within the high risk group that did not show up in the 12-lead resting ECGs (i.e. paroxysmal, persistent AF). However, the devices were not well tolerated by the participants and only 330 participants received them. No additional AF cases could be identified by the 24-hrs ECGs. Thus, this strategy was cancelled early on in the study and CHARGE-AF scores were not determined systematically. We amended the text in methods section accordingly. 

Add the CHARGE-AF score for each group of PD as a part of the results.

Reply: The scores are available only for a small fraction recruited at the very beginning. Thus, we refrain from reporting them.

The authors include the LVEF in different groups of PD as a part of their results. Do they have information of other echocardiographic parameters? It would be particularly interesting if they provide data on left atrium (volume, diameter), since it is a marker of FA risk.

Reply: These data are not yet fully quality controlled and could therefore not be used.

This is a suggestion: Most of the data that shows an association between PD and cardiac structural and functional abnormalities detected with echocardiogram (mass, myocardial deformation), comes from studies in patients with diabetes mellitus, hypertension and chronic kidney disease. In all of these studies, there was a clear association between PD severity and left ventricular hypertrophy and abnormal ventricular function. (You can see: Periodontal Disease, Systemic Inflammation and the Risk of Cardiovascular Disease. Heart Lung Circ. 2018 Nov;27(11):1327-1334, for a description of some of these studies). In our laboratory, we have done some work in otherwise healthy subjects with different grades of PD using echocardiography, and we could not find an association between PD and LVH or abnormalities in LV function. Although the reasons can be diverse, there is a possibility that PD and the systemic inflammatory process that comes with it acts as a boost for CVD; perhaps, PD might not be enough to cause CV abnormalities, but when added to other types of stress to the CV system (Hypertension, DM) it can cause CVD.

Reply: This is an interesting idea. We are constantly expanding the cohort to increase the case numbers and thus the study’s power. At present, the power is still not high enough for informative sub-group analyses of the kind you are suggesting. 

Since AF is multifactorial, and abnormalities in cardiac structure and function can play a role, this can be important for your study. More than 50 of your study population have hypertension; perhaps, you can consider a last analysis including only those with HTN and divide them in those with none/mild PD and severe PD to look for an association between AF and PD, in unadjusted and adjusted analysis.

Reply: We did this. Please, see Results section and supplementary material.

Reviewer #3:

Recent results from ARIC cohort study on PD-AF association not discussed

Reply: Okay, fixed.

DMFT assessed, a measure primarily for Caries. Data not used for analysis

Reply: We have not used the DMFT for any evaluations because teeth loss can result from various issues, such as trauma, orthodontic treatment or tooth decay. Nevertheless, the DMFT may be interesting for some readers because it is a widely used indicator of overall dental health. The actual values show that our cohort in fact has a very good overall oral health.

CRP and IL-6 are potential mediators, hence should not be adjusted for

Reply: Very good point. We performed a mediation analysis for CRP and IL-6. Please see Results section and supplementary material.

Socioeconomic status is a potential confounder that is not assessed

Reply: Thank you, that is true. We included educational level in the revision as a proxy for the SES (It is difficult to obtain correct income information in surveys, which renders the SES unreliable).

Based on hypothesis oral care should be associated with less AF

Reply: The prevalence of AF was lower in participants brushing their teeth 2x and more compared with those brushing less than 2 times daily. 

Assessment of AF using questionnaire not a validated method

Reply: Paroxysmal and some persistent forms of AF, which account for ~50% of all cases of AF, are difficult to detect by a single ECG examination in an epidemiological study. Therefore, we relied also on the medical history to identify such cases. People are usually aware of their diagnosis and about ¼ of participants who reported a history of AF could actually be verified by ECG in our study. In contrast, only 0.4% of those who reported no history of AF presented with AF according to the ECG. This shows that verifiable cases were highly enriched in the self-reporting group. Nevertheless you are right that self-reporting may have led to some miss-classification. On the other hand, we would have missed most cases of paroxysmal or persistent AF, if we had not included self-reported AF. We added this consideration to the discussion section.

---

## [Decision Letter · Decision Letter 2]

7 Oct 2021

PONE-D-21-08970R2Periodontitis, dental plaque, and atrial fibrillation in the Hamburg City Health StudyPLOS ONE

Dear Dr. Seedorf,

Thank you for submitting your manuscript to PLOS ONE. After careful consideration, we feel that it has merit but does not fully meet PLOS ONE’s publication criteria as it currently stands. Therefore, we invite you to submit a revised version of the manuscript that addresses the points raised during the review process.

Your revised paper was reevaluated by the previous reviewers. Though the quality of paper has been improved, one reviewer made some suggestions to further improve the analysis. Please read the comments carefully and address the issues accordingly. 

We look forward to receiving your revised manuscript.

Kind regards,

Tomohiko Ai, M.D., Ph.D.

Academic Editor

PLOS ONE

Journal Requirements:

Reviewers' comments:

Reviewer's Responses to Questions

**Comments to the Author**

1. If the authors have adequately addressed your comments raised in a previous round of review and you feel that this manuscript is now acceptable for publication, you may indicate that here to bypass the “Comments to the Author” section, enter your conflict of interest statement in the “Confidential to Editor” section, and submit your "Accept" recommendation.

Reviewer #1: (No Response)

Reviewer #2: All comments have been addressed

Reviewer #3: All comments have been addressed

2. Is the manuscript technically sound, and do the data support the conclusions?

Reviewer #1: Yes

Reviewer #2: Yes

Reviewer #3: Partly

3. Has the statistical analysis been performed appropriately and rigorously? 

Reviewer #1: Yes

Reviewer #2: Yes

Reviewer #3: Yes

4. Have the authors made all data underlying the findings in their manuscript fully available?

Reviewer #1: Yes

Reviewer #2: Yes

Reviewer #3: No

5. Is the manuscript presented in an intelligible fashion and written in standard English?

Reviewer #1: Yes

Reviewer #2: Yes

Reviewer #3: Yes

6. Review Comments to the Author

Reviewer #1: No further suggestions as the author has responded to all points and improved the overall validity of the article

Reviewer #2: The authors have made a great effort addressing the reviewers’ suggestions; the quality of the manuscript has improved.

Although the overall results of the study do not suggest an association between PD and AF, the study is still of relevance since it adds valuable information to the field, I think the paper is now ready to be considered for publication.

Reviewer #3: The reviewers comments have been addressed. I would suggest following changes:

1) Abstract conclusion starts with "no finding" first, and readers have to go sentences down to figure out what the positive findings of PI AF association

2) DMFT is an useful marker of oral health and dental caries. If the DMFT score is better in the group, please indicate that as a reason why it was not tested for association with AF. Alternately please include the analysis for readers to know that DMFT is not associated with AF

7. PLOS authors have the option to publish the peer review history of their article (what does this mean?). If published, this will include your full peer review and any attached files.

Reviewer #1: **Yes: **Amaar Obaid Hassan

Reviewer #2: **Yes: **Edgar Francisco Carrizales-Sepulveda

Reviewer #3: **Yes: **Souvik Sen MD

---

## [Author Response · Author response to Decision Letter 2]

15 Oct 2021

Please see uploaded file "Respons to reviewers"

---

## [Decision Letter · Decision Letter 3]

25 Oct 2021

Periodontitis, dental plaque, and atrial fibrillation in the Hamburg City Health Study

PONE-D-21-08970R3

Dear Dr. Seedorf,

We’re pleased to inform you that your manuscript has been judged scientifically suitable for publication and will be formally accepted for publication once it meets all outstanding technical requirements. I think this study will be an important reference regarding association between oral hygiene and AF. Congratulations!

Kind regards,

Tomohiko Ai, M.D., Ph.D.

Academic Editor

PLOS ONE

Additional Editor Comments (optional):

Reviewers' comments:

Reviewer's Responses to Questions

**Comments to the Author**

1. If the authors have adequately addressed your comments raised in a previous round of review and you feel that this manuscript is now acceptable for publication, you may indicate that here to bypass the “Comments to the Author” section, enter your conflict of interest statement in the “Confidential to Editor” section, and submit your "Accept" recommendation.

Reviewer #3: All comments have been addressed

2. Is the manuscript technically sound, and do the data support the conclusions?

Reviewer #3: Yes

3. Has the statistical analysis been performed appropriately and rigorously? 

Reviewer #3: Yes

4. Have the authors made all data underlying the findings in their manuscript fully available?

Reviewer #3: Yes

5. Is the manuscript presented in an intelligible fashion and written in standard English?

Reviewer #3: Yes

6. Review Comments to the Author

Reviewer #3: Thank you for responding to all comments. All comments have been addressed including addition of DMFT data.

7. PLOS authors have the option to publish the peer review history of their article (what does this mean?). If published, this will include your full peer review and any attached files.

Reviewer #3: **Yes: **Souvik Sen MD

---

## [Editor Report · Acceptance letter]

2 Nov 2021

PONE-D-21-08970R3 

Periodontitis, dental plaque, and atrial fibrillation in the Hamburg City Health Study 

Dear Dr. Aarabi:

I'm pleased to inform you that your manuscript has been deemed suitable for publication in PLOS ONE. Congratulations! Your manuscript is now with our production department. 

Kind regards, 

on behalf of

Dr. Tomohiko Ai 

Academic Editor

PLOS ONE